# Predicting the Epidemiological Dynamics of Lung Cancer in Japan

**DOI:** 10.3390/jcm8030326

**Published:** 2019-03-08

**Authors:** Takayuki Yamaguchi, Hiroshi Nishiura

**Affiliations:** 1Graduate School of Medicine, Hokkaido University, Kita 15 Jo Nishi 7 Chome, Kita-ku, Sapporo-shi, Hokkaido 060-8638, Japan; tyamaguchi@med.hokudai.ac.jp; 2CREST, Japan Science and Technology Agency, Honcho 4-1-8, Kawaguchi, Saitama 332-0012, Japan

**Keywords:** tobacco, lung neoplasms, forecasting, simulations, epidemiology, Japan

## Abstract

While the prevalence of smoking has steadily declined over time, the absolute numbers of lung cancer cases and deaths have continued to increase in Japan. We employed a simple mathematical model that describes the relationship between demographic dynamics and smoking prevalence to predict future epidemiological trends of lung cancer by age and sex. Never-smokers, smokers, and ex-smokers were assumed to experience different hazard of lung cancer, and the model was parameterized using data from 2014 and before, as learning data, and a future forecast was obtained from 2015 onwards. The maximum numbers of lung cancer cases and deaths in men will be 76,978 (95% confidence interval (CI): 76,630–77,253) and 63,284 (95% CI: 62,991–63507) in 2024, while those in women will be 42,838 (95% CI: 42,601–43,095) and 32,267 (95% CI: 32,063–32,460) in 2035 and 2036, respectively. Afterwards, the absolute numbers of cases and deaths are predicted to decrease monotonically. Our compartmental modeling approach is well suited to predicting lung cancer in Japan with dynamic ageing and drastic decline in smoking prevalence. The predicted burden is useful for anticipating demands for diagnosis, treatment, and care in the healthcare sector.

## 1. Introduction

In Japan, the absolute numbers of cases of and deaths from lung cancer have been increasing for decades (Figure 1A,B) [1]. The numbers of newly diagnosed cases of lung cancer in men and women increased from 12,701 in 1975 to 77,617 in 2014 and from 4971 in 1975 to 36,933 in 2014, respectively. The increase is considered to have reflected demographic ageing and past exposure history to smoking among older adults (Figure 1C,D) [1]. Similarly, the numbers of lung cancer deaths in men and women increased from 10,711 in 1975 to 52,505 in 2014 and from 4048 in 1975 to 20,891 in 2014, respectively [1,2]. If diagnosed with lymph node or distant metastasis, the effectiveness of treatment for lung cancer is known to be limited, and the overall five-year survival rate of patients diagnosed in 2006–2008 is as small as 31.9 % [2,3]. Pathologically, different types of lung cancers occur at different anatomical sites and have different characteristics. Squamous cell carcinoma and small cell carcinoma are common among men, especially smokers aged 35 years or older. They often arise centrally in larger bronchi, and small cell carcinoma is more rapidly metastatic than other types. Pulmonary adenocarcinoma is the most common type of lung cancer, seen in both smokers and lifelong nonsmokers, but most cases are associated with smoking, and it is common among women. These types of lung cancer share the common characteristic of an established causal link to tobacco smoking [3].

In Japan, the population dynamics of smokers have dramatically changed over time (Figure 1D) [4]. In 1965, the smoking prevalence was 82.3% and 15.7% in men and women, respectively, and these figures have dropped to 29.7% and 9.7% in 2016, respectively [4]. In addition, interventions against secondhand smoking, including separation of smoking areas from public spaces and a total ban on smoking within buildings, are underway. Because of these drastic declines in exposure over time, it is anticipated that the epidemiological dynamics of lung cancer will also change dramatically in the future, reflecting the dynamic decline in the population at risk. However, the epidemiological dynamics of lung cancer are also regulated by demographic dynamics: Japan is a superaged nation, with a dependency ratio that will exceed 50% in the near future (Figure 1C) [5,6]. As the country still has ultralow fertility, the ageing is still expected to progress [6]. Then, the incidence of lung cancer per 100,000 persons could continue to increase, fueled by the inflow of older people who become increasingly aged and have an elevated risk of contracting lung cancer. Besides, as Japan’s population has started to decrease, the prevalence of smoking has also monotonically decreased, and thus, it is anticipated that the absolute number of lung cancer cases will eventually stop growing.

Considering these two contradicting dynamic features, i.e., decreasing prevalence of smokers and ageing in declining population, predicting the epidemiological dynamics of lung cancer in Japan’s future require us to account for detailed underlying population dynamics, including time-dependent changes in the population at high risk of lung cancer. While qualitative discussions on the eventual decline of lung cancer have taken place elsewhere [7], and although various modeling studies have been conducted in North America to explore the impact of anti-smoking measures on lung cancer epidemiology [8,9], we have yet to capture the dynamics in Japan or to fully grasp the outcomes using a mathematical model that explicitly describes the underlying population dynamics.

The purpose of the present study is to devise a simple mathematical model that allows us to describe both demographic dynamics and prevalence of smoking to appropriately predict future epidemiological trends of lung cancer by age and sex in Japan.

## 2. Materials and Methods

### 2.1. Epidemiological and Demographic Data

To accomplish the goal of this study, we reconstructed the demographic and epidemiological dynamics of smokers and patients with lung cancer. To do so, we extracted five pieces of information as the input data: (i) lung cancer incidence, (ii) lung cancer mortality, (iii) smoking prevalence, (iv) age- and sex-specific population size, and (v) mortality in general. For (i), we used the Cancer Registry and Statistics from the Cancer Information Service, National Cancer Center, Japan by 5-year age groups from 0 to 85 years and by sex from 1975 to 2014 [1]. The nationwide cancer registry system was fully set up in Japan only recently, and for this reason, the estimated cancer incidence from 32 population-based cancer registries was used as a substitute to represent the incidence of lung cancer in Japan [10]. For (ii) and (v), we used cause-specific mortality, given as the yearly number of deaths by cause, by 5-year age groups and sex from 1965 to 2014, from data publicly available from the Health, Labour and Welfare Statistics Association [5,11]. The time series of smoking prevalence (i.e., dataset (iii)) from 1965 to 2016 was derived from the report by Japan Tobacco, Inc. by 10-year age groups from 20 to 60 years and by sex [4,12]. Although the national survey (i.e., the National Health and Nutrition Survey in Japan) also recorded similar datasets of smoking prevalence, the national survey data have only been available since 1989, and because of the overall similarity between the two datasets, we decided to use the Japan Tobacco dataset on account of its longer time series. Moreover, we rigorously fitted our model to the age- and sex-specific population sizes in Japan (i.e., dataset (iv)) using census data from the Statistics Bureau, Ministry of Internal Affairs and Communications, Japan to obtain the population size by age (0–100 years) and sex [13]. Because the census is taken only every 5 years, and the population size in gap years was extrapolated by the Ministry of Health, Labour and Welfare, we used the compensated population estimates for every year. In addition, future population sizes from 2016 to 2065 were used to obtain future predictions of lung cancer, and for this purpose, the projected population by age and sex was extracted from the predictions of the National Institute of Population and Social Security, Japan [6].

### 2.2. Mathematical Model

We used a compartmental mathematical model that describes the natural history of smoking and the differential hazard rates of developing lung cancer according to different smoking statuses (Figure 2). Every person is born as a never-smoker, but people experience the hazard of becoming a smoker that depends on time and age. Smokers can quit smoking during the course of life to become ex-smokers. Let t and a represent time and age, respectively. The population’s smoking status is expressed by M(t,a), S(t,a), and E(t,a), representing the number of never-smokers, smokers, and ex-smokers, respectively, in year *t* at age *a*. Never-smokers begin smoking at a rate of σ(t,a), and smokers cease smoking at a rate of δ(t,a). For simplicity, we assume that ex-smokers do not restart smoking again (i.e., we assume that purported ex-smokers who failed to quit smoking for a long period were not truly ex-smokers and that they remained in the smokers’ compartment). All three compartments experience two different hazards: the hazard of lung cancer and the mortality hazard due to causes other than lung cancer. The hazard of lung cancer among never-smokers is time- and age-dependent, λ(t,a), and smokers and ex-smokers experience $k_{S}$ and $k_{E}$ times greater hazard rates of lung cancer, which causes the individual enter the compartment of patients with lung cancer L(t,a), who die at a rate of μ(t,a)+ν(t,a), where μ(t,a) is the mortality rate of the general population due to any cause other than lung cancer and ν(t,a) is the excess mortality due to lung cancer. To represent the elevated risk of death by smoking status, we use $1+q_{S}$ and $1+q_{E}$ as the hazard ratios of death from any cause other than lung cancer among smokers and ex-smokers, respectively, as compared with never-smokers. That is, never smokers, smokers, and ex-smokers die at the rates μ(a), $\left(1+q_{S}\right)\mu \left(a\right)$, and $\left(1+q_{E}\right)\mu \left(a\right)$, respectively.

The dynamics is governed by the following partial differential equations:(1)$$\begin{array}{l} \left(\frac{\partial }{\partial t}+\frac{\partial }{\partial a}\right)M\left(t,a\right)=-\left(\mu \left(t,a\right)+\sigma \left(t,a\right)+\lambda \left(t,a\right)\right)M\left(t,a\right),\\ \left(\frac{\partial }{\partial t}+\frac{\partial }{\partial a}\right)S\left(t,a\right)=\sigma \left(t,a\right)M\left(t,a\right)-\left(\left(1+q_{S}\right)\mu \left(t,a\right)+\delta \left(t,a\right)+k_{S}\lambda \left(t,a\right)\right)S\left(t,a\right),\\ \left(\frac{\partial }{\partial t}+\frac{\partial }{\partial a}\right)E\left(t,a\right)=\delta \left(t,a\right)S\left(t,a\right)-\left(\left(1+q_{E}\right)\mu \left(t,a\right)+k_{E}\lambda \left(t,a\right)\right)E\left(t,a\right),\\ \left(\frac{\partial }{\partial t}+\frac{\partial }{\partial a}\right)L\left(t,a\right)=\lambda \left(t,a\right)\left(M\left(t,a\right)+k_{S}S\left(t,a\right)+k_{E}E\left(t,a\right)\right)-\left(\mu \left(t,a\right)+\nu \left(t,a\right)\right)E\left(t,a\right), \end{array}$$
with the initial conditions and the boundary conditions
$$\begin{array}{l} M\left(t_{0},a\right)=M_{0}\left(a\right),\\ S\left(t_{0},a\right)=S_{0}\left(a\right),\\ E\left(t_{0},a\right)=E_{0}\left(a\right),\\ L\left(t_{0},a\right)=L_{0}\left(a\right),\\ M\left(t,a_{0}\right)=B\left(t\right),\\ S\left(t,a_{0}\right)=0,\\ E\left(t,a_{0}\right)=0,\\ L\left(t,a_{0}\right)=0. \end{array}$$
where $t_{0}$ is the initial time, $a_{0}$ is the youngest age of the target population, B(t) is the number of population members aged $a_{0}$ at time t, and $M_{0}\left(a\right)$, $S_{0}\left(a\right)$, $E_{0}\left(a\right)$, and $L_{0}\left(a\right)$ are the initial age distributions of the population. To fit the continuous model to discrete data, the abovementioned model was rewritten as ordinary differential equations, the details of which are available in the Appendix.

### 2.3. Statistical Estimation

We quantified the system (1) without imposing excessively unnecessary assumptions. Only parameters $k_{S}$, $k_{E}$, $q_{S}$, and $q_{E}$, i.e., the hazard ratios of lung cancer among smokers and ex-smokers and excess hazard ratios of mortality for reasons other than lung cancer among smokers and ex-smokers, respectively, were extracted from the literature and fixed: $k_{S}=4.94$, $k_{E}=2.20$, $q_{S}=0.49$, $q_{E}=0.20$ for men and $k_{S}=4.25$, $k_{E}=2.19$, $q_{S}=0.51$, $q_{E}=0.46$ for women [14]. Other published studies from Japan yielded similar estimates [15,16,17,18,19]. All remaining rates were estimated via the following fitting procedure. As mentioned above, we used five different datasets, including the (i) incidence, (ii) mortality of lung cancer, (iii) smoking prevalence, (iv) population size, and (v) mortality other than lung cancer. Smoking prevalence was assumed to have been the result of binomial sampling, while other data-generating processes were each approximated as Poisson processes (see the Appendix for a detailed description of the likelihood function). We employed the maximum likelihood method to estimate parameters and derived the 95% confidence intervals (Cis) of model estimates (e.g., intervals of projected numbers of patients with lung cancer) using a parametric bootstrap method. During the bootstrapping exercise, model parameters were resampled from a multivariate normal distribution with mean $\bar{\theta }$ and variance-covariance matrix Σ, where $\bar{\theta }$ is the estimate obtained by maximum likelihood estimation and Σ is the inverse of the Hessian matrix of −logL(θ) at $\theta =\bar{\theta }$. We obtained the model solution for each set of parameters and performed 1000 parameter resampling simulations to obtain a distribution for each model solution. We then computed the 95% CI by taking the 2.5th and 97.5th percentiles of the simulated distribution.

### 2.4. Ethical Considerations

The present study analyzed data that are publicly available. As such, the datasets used in our study were de-identified and fully anonymized in advance, and the analysis of publicly available data without identity information does not require ethical approval.

### 2.5. Data Sharing Policy

The major data source (i.e., the datasets on lung cancer incidence and mortality) is available in references [1,2] via the linked URLs.

## 3. Results

Figure 3 compares the observed and predicted incidence and mortality rates of lung cancer per 100,000 population along with forecasts for 2015–2050. Incidence and mortality were forecasted jointly by age and sex. Observed patterns were qualitatively captured very well for all the observed rates, indicating that both incidence and mortality rates per 100,000 population for the entire population as well as by gender will continue to steadily grow overall (also, see Appendix Figure A1). However, the incidence and mortality of older men aged 75 years or older are expected to have already peaked by 2015 (Figure 3A,C) and are predicted to decrease over time, whereas those among women will steadily increase over time in the future (Figure 3B,D). The predicted incidence and mortality among men aged 65–74 years will slightly decrease with time in men and will probably peak around 2045 in women. Long-term decline in the incidence and mortality in men’s age groups other than those aged 75 years or older is smaller, because there is ageing within age groups, and thus, the risk of lung cancer is elevated over time within the same age group.

When the absolute numbers of lung cancer cases and deaths are examined by age group and sex (Figure 4), the predicted patterns are more counterintuitive than those of the rates per 100,000 population. The number of new lung cancer cases will start to wane starting in the mid-2020s in men and around 2035 in women (Figure 4A,B), and these overall patterns also hold for the predicted numbers of lung cancer deaths (Figure 4C,D). The maximum numbers of new cases and deaths due to lung cancer in men will be 76,978 (95% CI: 76,630–77,253) and 63,284 (95% CI: 62,991–63507) in 2024, while those in women will be 42,838 (95% CI: 42,601–43,095) and 32,267 (95% CI: 32,063–32,460) in 2035 and 2036, respectively. The absolute numbers of lung cancer cases and deaths among older adults aged 65–74 years and 75 years or older will both fluctuate because of the underlying demographic dynamics (i.e., the visible impact of the baby boom in Japan), and for both men and women, the predicted numbers of cases and deaths will have mostly plateaued by 2030. Reflecting a decreased population and its overall ageing, the numbers of lung cancer cases and deaths among those aged 15–64 years will steadily decrease among both men and women (see Appendix Figure A2).

Figure 5 shows the age-dependent dynamics and forecast of smoking prevalence. Smoking prevalence has drastically decreased in men over time, and the declining trend in the working age population is expected to continue until around 2030 (Figure 5A). Because of Japan’s quickening rate of demographic decline, the decrease in smoking prevalence is expected to plateau among those aged 15–64 years. The prevalence in women is lower than that of men, and women’s overall prevalence of smoking should eventually be close to or slightly lower than that of men (Figure 5B). As time passes, in terms of prevalence, ex-smokers will increasingly dominate the older adult population (Figure 5C,D). Especially, among men, more than 50% were predicted to be ex-smokers by 2015, and this increase will continue even after 2030. A similar increasing trend of ex-smokers is also expected among women by 2030.

Because the hazard ratio $k_{s}$ of lung cancer among smokers compared with never-smokers is one of the most important but uncertain quantities that could influence our forecast, sensitivity analysis of the predicted values was conducted by varying the relative value of $k_{s}$ (Figure 6). When we varied the hazard ratio from 0.5 to 1.5 times the original value, both lung cancer incidence and mortality per 100,000 population changed only slightly. Among men, the incidence and mortality increased with 1.5 times the original *k*_s_, but we did not observe any abrupt decline by employing 0.5 times $k_{s}$, perhaps because of the greater impact of smokers and ex-smokers on the demographic dynamics of the population of patients with lung cancer (Figure 6A). Among women, a monotonic increase in both incidence and mortality was observed when the multiplier of the hazard ratio was increased from 0.5 to 1.5 (Figure 6B).

## 4. Discussion

The present study parameterized a mathematical model that governs the demographic dynamics and population dynamics of smokers, allowing us to examine the time-dependent epidemiological dynamics of lung cancer in the future of Japan. The model was parameterized using data from 2014 and before as learning data and offered future predictions of lung cancer incidence and deaths, showing that the absolute numbers of new cases and deaths from lung cancer will increase until the mid-2020s in men and mid-2030s in women, respectively. Afterwards, the absolute numbers of cases and deaths were predicted to decrease monotonically, reflecting the decline in smoking prevalence and future population decline in Japan [20]. However, the incidence and mortality predictions per 100,000 persons continued to increase slightly, contradicting the overall decreasing trends in Western countries. The predicted burden of lung cancer is likely useful for anticipating future demands for diagnosis, treatment, and care in the healthcare sector. To our knowledge, except for individual risk-based prediction using cohort epidemiological data [21], the present study is the first to have offered an explicit prediction of lung cancer rates at a population level in Japan.

As an important lesson, we have successfully quantified the impact of decline in smoking rate and decreasing population on the epidemiology of lung cancer in an objective manner. Forecasted absolute numbers of cases and deaths are critical to plan future healthcare services, including estimations of required medical services (e.g., demand for screening) and hospital beds at facilities. In particular, we have shown that the dramatic increase in demand will peak in the mid-2030s among women, and subsequently, the demand is expected to wane drastically because of a decreasing population at high risk of lung cancer. Rather than the absolute value, the overall social burden of lung cancer might be better measured by the rate per 100,000 population. It is critical that the rate is expected to increase steadily over time, although breakthroughs in prevention or treatment of lung cancer might occur in the future. Our finding contradicts the ongoing epidemiological trends in other industrialized countries where both the incidence and mortality are in declining trend: the impact of rapid ageing has been very influential in Japan. The good news is that the absolute numbers of cases and deaths are expected to start to wane from the 2030s, and Japan may have already experienced its peak incidence rate of lung cancer in older men. Different trends between the absolute number and the estimate per 100,000 indicate a critical importance to account for the demographic dynamics (e.g., a flat or slight increase in the incidence among elderly aged 75 years or older in Figure 3C).

In addition to the value of overall prediction of epidemiological trends and medical demand related to lung cancer, the use of the proposed mathematical model has two remarkable advantages. First, our model appropriately offers predictions by age and sex, and thus would be useful for anticipating demands for diagnosis, treatment, and care. For instance, one of our long-term forecasts indicates that the absolute numbers of cases and deaths among those aged 75 years or older will increase again starting in the 2040s, while those among the working-age population will continue to decrease steadily (Figure 4). This is alarming for the treatment of patients over the next 30 years. Such fluctuating patterns were appropriately captured because our predictions were firmly supported by the tightly predicted demographic dynamics, including ageing, and this feature is well suited to the Japanese setting with its fast demographic changes over time. Second, the present study successfully obtained future trends of smoking prevalence using a simplistic structure of the natural history of smoking. Reflecting the trend of smoking prevalence, the decline in absolute number of cases in women was shown to be smaller than that in men (Figure 4). Moreover, smoking prevalence was firmly predicted to decrease monotonically in the future. However, the prevalence of ex-smokers will continue to grow, and this will be a remarkable characteristic among older adults in the future, putting a substantial number of people at risk of lung cancer.

By employing the proposed modeling scheme, public health insights into the effectiveness of anti-smoking measures could potentially be gained while appropriately accounting for the underlying demographic dynamics and smoking prevalence. Published modeling methods that employ statistical modeling techniques, including survival analysis and individual-based (micro)simulations, can also be used for similar assessments [8,9,22,23,24,25,26,27,28,29,30,31,32,33,34,35,36,37,38,39,40,41,42,43,44,45,46], but manual control of the population at risk is required in these simplistic approaches. Our proposed model does not have to involve such manual input of the population size at risk because we explicitly model the demographic dynamics and examine the future course of smoking prevalence within our model directly when predicting the numbers of lung cancer cases [25,26,28,31,32,45]. Another direction of possible application would be to explore the effectiveness of additional screening, including regular or non-regular screening with chest X-rays or computed tomography, which is our area of ongoing future study [27,29,30,31,37,42,44].

While our model appears to be practically useful, the proposed modeling study involved a number of limitations that must be described here. First, our model does not account for future improvement in prevention and treatment. Presently, lung cancer remains one of the leading causes of death (i.e., it has a severe prognosis), but gene therapy for lung cancer and other novel approaches could drastically change case outcomes in the future. Second, risk factors other than smoking were not explicitly considered. Genetic predisposition and well-known carcinogenic exposures could be used to attain earlier diagnosis and improve prognosis [47,48]. In fact, the relative risk of smoking in causing lung cancer among Japanese is known as smaller than those in the United States and Europe, referred to as the smoking paradox [49]. Third, we did not have substantial datasets to incorporate the impact of smoking dose (e.g., pack-years) into our model, although the dose is known to be influential in clinical settings. Fourth, we ignored the impact of secondhand smoking on lung cancer, while it has been documented that secondhand exposure in households acts as a cause [50]. Incorporating secondhand smoking might require modeling and data at greater spatial precision (e.g., household-level data). Fifth, to obtain future forecasts, we have used the projected population in Japan and ignored the mortality rate and other demographic parameters [51]. If a drastic change happens in other diseases (e.g., certain types of cancers become not lethal at all someday) [52], the epidemiological structure of mortality would change and mortality due to lung cancer would also be influenced. Lastly, it has been pointed out that early initiation of smoking has become more and more common in Japan [20]. Although our model accounted for time-dependent hazard rate to start smoking, the future prediction had to assume a time-dependent hazard which was derived from the most recent observed time [53,54,55].

Despite these limitations, we believe that the present study successfully captured the demographic and population dynamics of smokers in Japan, yielding appropriate estimates and forecasts of lung cancer. We are open to criticisms or comments for future collaborations, hoping to expand the proposed modeling schemes to other types of cancer and extend the lung cancer model to scenario analysis of interventions.

## 5. Conclusions

The present study parameterized a mathematical model that governs the demographic and population dynamics of smokers, predicting the epidemiological dynamics of lung cancer in Japan. The model was parameterized using data from 2014 and before as learning data, and offered future predictions of lung cancer incidence and deaths, showing that the absolute numbers of new lung cancer cases and deaths will increase until the mid-2020s in men and the mid-2030s in women. The model can potentially be further applied to elucidate the effectiveness of anti-smoking measures and screening procedures at reducing lung cancer mortality.

## Figures and Tables

**Figure 1 jcm-08-00326-f001:**
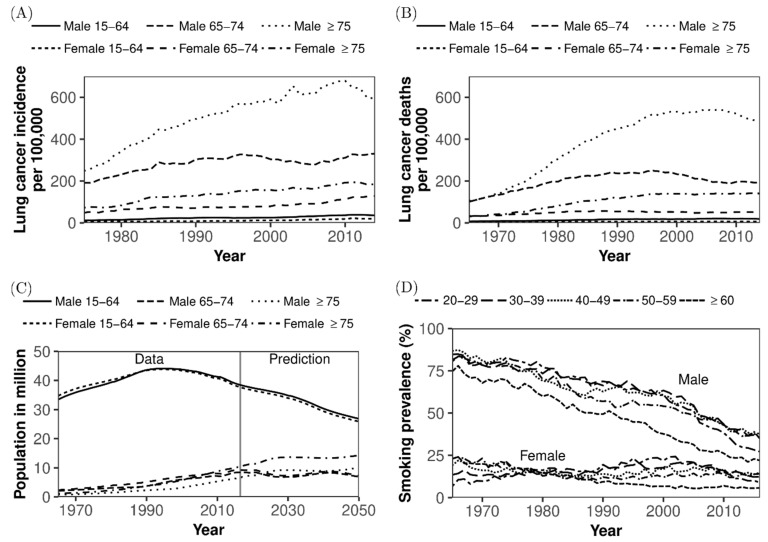
Incidence and mortality of lung cancer and the demographic dynamics and smoking prevalence in Japan. Published data of lung cancer incidence and mortality and the demographic dynamics and smoking prevalence in Japan. (**A**) Incidence of lung cancer by age group and sex per 100,000 population, (**B**) Mortality due to lung cancer by age group and sex per 100,000, (**C**) Age distribution of the population by sex, and (**D**) Prevalence of smokers by age group and sex.

**Figure 2 jcm-08-00326-f002:**
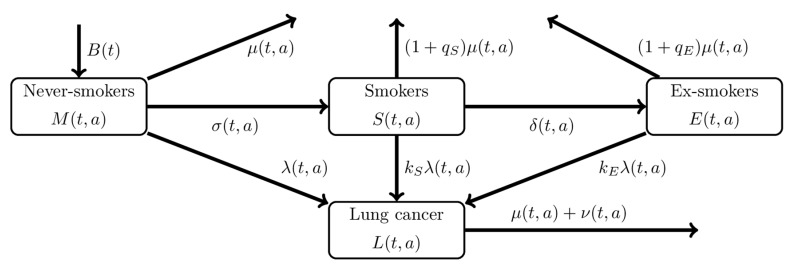
Compartmental model of lung cancer. Compartment diagram of the proposed model to predict lung cancer from the epidemiological, demographic, and smoking data. The population is categorized by smoking and lung cancer status. Never smokers begin smoking at a transition rate σ and move to the compartment of smokers. Smokers cease smoking at a transition rate δ and become ex-smokers. For simplicity and because of the absence of data over long time series, we assume that ex-smokers do not restart smoking. Never-smokers, smokers, and ex-smokers experience different hazard rates of lung cancer at λ, $k_{S}\lambda$, and $k_{E}\lambda$, respectively. Members of the population in each compartment die at given mortality rates.

**Figure 3 jcm-08-00326-f003:**
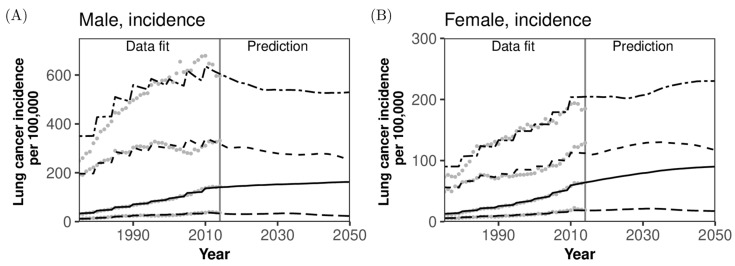
Comparison between observed and predicted lung cancer incidence and mortality per 100,000 population in Japan. Comparisons between observed and predicted lung cancer incidence and mortality per 100,000 population are made by age group and sex. (**A**) Lung cancer incidence in men, (**B**) Lung cancer incidence in women, (**C**) Mortality in men, and (**D**) Mortality in women. Bold dashed line represents working-age population aged 15–64 years, thin dashed line corresponds to those aged 65–74 years, uneven dashed line represents those aged 75 years or older, and solid line represents the entire population. Grey dots before 2014 were used to estimate model fitting parameters via the maximum likelihood method, and lines for 2015 and later are predictions calculated using maximum likelihood estimates at the latest time. The 95% confidence intervals are omitted from predictions because of the narrow uncertainty bounds.

**Figure 4 jcm-08-00326-f004:**
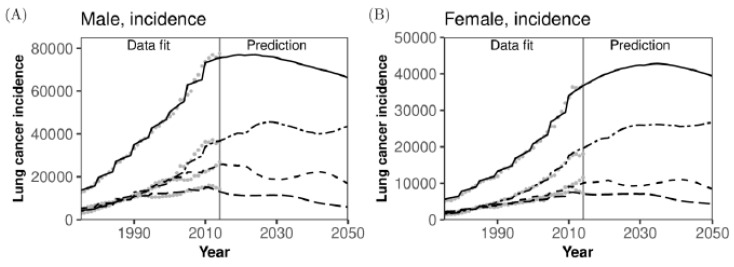
Comparisons between observed and predicted absolute numbers of lung cancer cases and deaths in Japan. Comparisons between observed and predicted absolute numbers of lung cancer cases and deaths are shown by age group and sex. (**A**) Lung cancer incidence in men, (**B**) Lung cancer incidence in women, (**C**) Mortality in men, and (**D**) Mortality in women. Bold dashed line represents working-age population aged 15–64 years, thin dashed line corresponds to those aged 65–74 years, uneven dashed line represents those aged 75 years or older, and solid line represents the entire population. Grey dots before 2014 were used to estimate model fitting parameters via the maximum likelihood method, and lines for 2015 and later indicate predictions calculated using maximum likelihood estimates at the latest time. The 95% confidence intervals are omitted from predictions because of the narrow uncertainty bounds.

**Figure 5 jcm-08-00326-f005:**
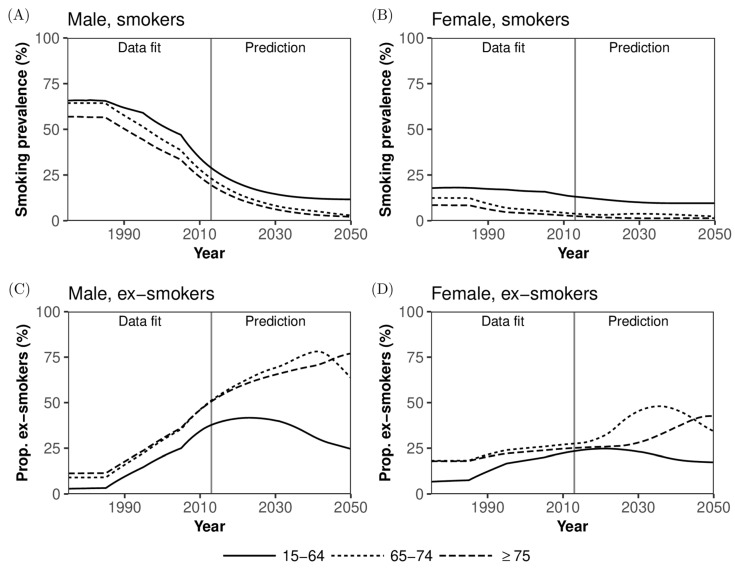
Smoking prevalence in Japan. Comparisons between observed and predicted smoking prevalence in Japan. (**A**) Smoking prevalence in men, (**B**) Smoking prevalence in women, (**C**) Ex-smokers’ prevalence in men, and (**D**) Ex-smokers’ prevalence in women. Solid line represents the estimated prevalence among those aged 15–64 years, dotted line represents the prevalence among those aged 65–74 years, and dashed line shows the estimate among people aged 75 years or older. Lines for 2015 and later indicate predictions calculated using maximum likelihood estimates at the latest time.

**Figure 6 jcm-08-00326-f006:**
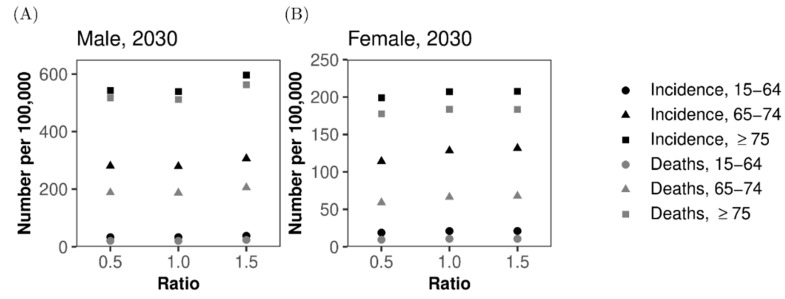
Sensitivity of lung cancer incidence and mortality to the relative hazard ratio among smokers. Predicted incidence and mortality of lung cancer are examined according to the different relative hazard ratios of lung cancer among smokers. (**A**) Incidence and mortality in men and (**B**) Incidence and mortality in women. Black marks show the incidence, while grey marks represent mortality. Circles, triangles, and squares represent those aged 15–64 years, 65–74 years, and 75 years or older, respectively.

## Data Availability

Collected cancer datasets are publicly available and can be retrieved from references [1,2].

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
