# Peer review of "Predicting the Epidemiological Dynamics of Lung Cancer in Japan"

_jcm, 2019, doi:10.3390/jcm8030326_

Reviewer 1 Report

Epidemiological literature assumes that lung cancer before the age of 35 is not associated with smoking. What is the justification therefore to analyze the group from 15 years of age?  I suggest restriction to groups 35-64, 65-74 and 75+

All Figures concerning age groups should have a logarithmic scale, at the arithmetic the drawings are illegible

I do not understand why, with long-term decline in the smoking rate among men, lung cancer effect is visible only in the oldest group (> 75). In the remaining groups, the assessment based on the drawing was impossible.

In VHD Index countries, a decrease in the morbidity and mortality of men with lung cancer is observed. This article suggests growth in Japan in the future. This conclusion is in contradiction with the observed decline in the frequency of smoking. Such results require a detailed discussion along with the potential causes of such patterns.

The results obtained suggest that perhaps hazard ratios obtained from the study [14] is flawed. I would suggest a discussion of the results from the study [14] with those obtained in CPS I and II and achived for European populatios.

 It is important to acknowledge that the paper is useful for anticipating and planning demands for diagnosis, treatment, and care. Also for this purpose, in the Discussion section, authors stated that “the good news is that the incidence and mortality per 100,000 persons in men are expected to wane over time, and Japan may have already experienced its peak incidence rate of lung cancer in men [20].” (lines 277–279). Since the reference no. 20 is not a predictive analysis, the reader might guess this statement is in line with the results of the current analysis. At least, any contradiction is not marked in the Discussion section. Note, the results in Figure 3 indicate the opposite (total figures). Both total lung cancer incidence and mortality per 100,000 men are expected to increase over time. This might be puzzling for the policy maker and needs to be addressed in the Discussion section or in the rebuttal letter.

 Authors name some limitations of the study (lines 307–322). As suggested by Funatogawa, Funatogawa, and Yano (2013), early initiation of smoking is becoming more and more common in both sexes. In this context, authors might like to address the assumptions of the model (including the assumed stability of parameters over time).

 Some statements are inaccurate, e.g.: “the incidence and mortality predictions per 100,000 persons continued to increase slightly or remain almost a constant because these rates are less affected by the decreasing population size” (lines 261–263).

Predicted lung cancer mortality rate in men aged 75+ is initially flat or increasing, as shown in Figure 3C. Since Authors have underlined the importance of the study for policy makers, they might want to discuss in the Discussion section or in the rebuttal letter whether this is a further continuation of (counterfactual) increase in modelled lung cancer mortality rate in 2010–2014 or the binding information for policy enforcement.

Author Response

Point-by-point responses: Predicting the epidemiological dynamics of lung cancer in Japan (jcm-449169)

 Reviewer 1: Epidemiological literature assumes that lung cancer before the age of 35 is not associated with smoking. What is the justification therefore to analyze the group from 15 years of age? I suggest restriction to groups 35-64, 65-74 and 75+

>> 

We thank the reviewer for this comment. We have checked literature along with consultations on this matter with a cancer epidemiologist who let us know that teenagers and 20s are not included due to perceived small risk. In our study, there has been very few cases aged below 35 years and the impact of excluding 15-34 years has been minimal. We added a note that the smoking as a risk factor is expecially the case among adults aged 35 years or older (P1L36-37).

 All Figures concerning age groups should have a logarithmic scale, at the arithmetic the drawings are illegible

>> 

We apologize for this confusion. We have prepared logarithmic scale ones, but these do not allow identification of peak year in the absolute number of cases. Thus, we decided to add those logarithmic figures as the Appendix of this paper (P18).

 I do not understand why, with long-term decline in the smoking rate among men, lung cancer effect is visible only in the oldest group (> 75). In the remaining groups, the assessment based on the drawing was impossible.

>> 

Ageing occurs within the same age group, and that has prevented the incidence and mortality to abruptly decline. This was mentioned in P5L180-182.

 In VHD Index countries, a decrease in the morbidity and mortality of men with lung cancer is observed. This article suggests growth in Japan in the future. This conclusion is in contradiction with the observed decline in the frequency of smoking. Such results require a detailed discussion along with the potential causes of such patterns.

>> 

The increase in the total is likely because of demography. We have added discussion to P9L279-281.

 The results obtained suggest that perhaps hazard ratios obtained from the study [14] is flawed. I would suggest a discussion of the results from the study [14] with those obtained in CPS I and II and achived for European populatios.

>> 

We agree with the view of the reviewer. It has been referred to as the smoking paradox, and the hazard rate of smoking in causing lung cancer has been known as smaller among Japanese and Korean as compared with those in the Untied States and European countries. We added a short note referring to this matter (P10L317-319).

 It is important to acknowledge that the paper is useful for anticipating and planning demands for diagnosis, treatment, and care. Also for this purpose, in the Discussion section, authors stated that “the good news is that the incidence and mortality per 100,000 persons in men are expected to wane over time, and Japan may have already experienced its peak incidence rate of lung cancer in men [20].” (lines 277–279). Since the reference no. 20 is not a predictive analysis, the reader might guess this statement is in line with the results of the current analysis. At least, any contradiction is not marked in the Discussion section. Note, the results in Figure 3 indicate the opposite (total figures). Both total lung cancer incidence and mortality per 100,000 men are expected to increase over time. This might be puzzling for the policy maker and needs to be addressed in the Discussion section or in the rebuttal letter.

>> 

We added a clear mention about the usefulness for anticipating demands for diagnosis, treatment and care in P9L286-287. Reference [20] was removed from the corresponding sentence in P9. We have also emphasized that the increase of the total of incidence and mortality in men contradict the decreasing trends in Western countries (P9L264-265).

 Authors name some limitations of the study (lines 307–322). As suggested by Funatogawa, Funatogawa, and Yano (2013), early initiation of smoking is becoming more and more common in both sexes. In this context, authors might like to address the assumptions of the model (including the assumed stability of parameters over time).

>> 

We mentioned this issue in P10L328-332.

 Some statements are inaccurate, e.g.: “the incidence and mortality predictions per 100,000 persons continued to increase slightly or remain almost a constant because these rates are less affected by the decreasing population size” (lines 261–263).

>> 

The corresponding sentence was corrected (P9L264-265).

 Predicted lung cancer mortality rate in men aged 75+ is initially flat or increasing, as shown in Figure 3C. Since Authors have underlined the importance of the study for policy makers, they might want to discuss in the Discussion section or in the rebuttal letter whether this is a further continuation of (counterfactual) increase in modelled lung cancer mortality rate in 2010–2014 or the binding information for policy enforcement.

>> 

This point is explainable by ageing (i.e. underlying demographic dynamics) and the importance of this point was emphasized in P9L283-285.

Reviewer 2 Report

The  manuscript by Yamaguchi and Nishiura examines the future epidemiology of lung cancer in Japan using mathematical modeling.  The study is important as analyzing cancer incidence and mortality will assist in planning for appropriate resource and financial designation.  A recent analysis of cancer projections, in the setting of public health programs to decrease smoking rates in Japan has not been performed.  Their results show that even with restrictions on public smoking and health care programs to limit smoking that the incidence of lung cancer and lung cancer associated deaths for women will continue to increase as far as the projection period of 2050.  For men the findings are somewhat similar in that the incidence of lung cancer and cancer associated deaths will only decrease by a small amount out to 2050.  These findings are alarming and suggest that ongoing resources will need to be invested in Japan for the treatment of these patients over the next 30 years. This information will be extremely helpful to government planners to ensure that adequate resources are in place.  The limitations of the study are clearly stated including the possibility of newer diagnostic and treatment tools that could decrease mortality, the lack of ability to control for other risk factors that may have changed over the ensuing years and the inability to stratify rsik based on the amount of smoking (e.g. pack years).    

Author Response

Point-by-point responses: Predicting the epidemiological dynamics of lung cancer in Japan (jcm-449169)

 Reviewer 2: The manuscript by Yamaguchi and Nishiura examines the future epidemiology of lung cancer in Japan using mathematical modeling. The study is important as analyzing cancer incidence and mortality will assist in planning for appropriate resource and financial designation. A recent analysis of cancer projections, in the setting of public health programs to decrease smoking rates in Japan has not been performed. Their results show that even with restrictions on public smoking and health care programs to limit smoking that the incidence of lung cancer and lung cancer associated deaths for women will continue to increase as far as the projection period of 2050. For men the findings are somewhat similar in that the incidence of lung cancer and cancer associated deaths will only decrease by a small amount out to 2050. These findings are alarming and suggest that ongoing resources will need to be invested in Japan for the treatment of these patients over the next 30 years. This information will be extremely helpful to government planners to ensure that adequate resources are in place. The limitations of the study are clearly stated including the possibility of newer diagnostic and treatment tools that could decrease mortality, the lack of ability to control for other risk factors that may have changed over the ensuing years and the inability to stratify rsik based on the amount of smoking (e.g. pack years).

>> 

We appreciate the reviewer for positively assessing our manuscript. We have emphasized even more the implication of the need for treatment in P9L292.